# Dynamic Semi-Supervised Federated Learning Fault Diagnosis Method Based on an Attention Mechanism

**DOI:** 10.3390/e25101470

**Published:** 2023-10-21

**Authors:** Shun Liu, Funa Zhou, Shanjie Tang, Xiong Hu, Chaoge Wang, Tianzhen Wang

**Affiliations:** School of Logistic Engineering, Shanghai Maritime University, Shanghai 201306, China; 202130210112@stu.shmtu.edu.cn (S.L.); huxiong@shmtu.edu.cn (X.H.); cgwang@shmtu.edu.cn (C.W.); tzwang@shmtu.edu.cn (T.W.)

**Keywords:** fault diagnosis, semi-supervised federated learning, attention mechanism, dynamic federation, model reliability

## Abstract

In cases where a client suffers from completely unlabeled data, unsupervised learning has difficulty achieving an accurate fault diagnosis. Semi-supervised federated learning with the ability for interaction between a labeled client and an unlabeled client has been developed to overcome this difficulty. However, the existing semi-supervised federated learning methods may lead to a negative transfer problem since they fail to filter out unreliable model information from the unlabeled client. Therefore, in this study, a dynamic semi-supervised federated learning fault diagnosis method with an attention mechanism (SSFL-ATT) is proposed to prevent the federation model from experiencing negative transfer. A federation strategy driven by an attention mechanism was designed to filter out the unreliable information hidden in the local model. SSFL-ATT can ensure the federation model’s performance as well as render the unlabeled client capable of fault classification. In cases where there is an unlabeled client, compared to the existing semi-supervised federated learning methods, SSFL-ATT can achieve increments of 9.06% and 12.53% in fault diagnosis accuracy when datasets provided by Case Western Reserve University and Shanghai Maritime University, respectively, are used for verification.

## 1. Introduction

As an important part of modern industrial systems, the fault diagnosis of a rolling bearing is crucial [1,2,3]. Data-driven fault diagnosis methods can extract fault features directly from massive collections of data and allow for the construction of a fault diagnosis model for rapid equipment monitoring [4,5]. As a data-driven method, deep learning is more powerful in terms of representing complex nonlinear mapping relationships, so fault diagnosis methods based on deep learning are becoming more and more widely applied [6]. In practical industrial applications, the process of labeling a large quantity of data often demands significant human and material resources. Therefore, the construction of fault diagnosis models utilizing extensive unlabeled data has received a copious amount of attention from academics and industry experts [7,8]. Although unsupervised methodologies can solve the issue of unlabeled data, establishing a link between input data and output results is challenging due to the lack of known labels [9,10]. On the other hand, semi-supervised deep learning methods can optimize fault diagnosis models developed with minimal labeled data by utilizing a large quantity of unlabeled data, offering important engineering significance [11,12]. However, the pressing issue lies in how semi-supervised learning can be implemented in a single-client setting where no labeled data are present. Semi-supervised federated learning offers a solution, enabling clients bereft of labeled data to fortify their classification capabilities using information from other clients possessing labeled data. This response carries substantial engineering significance in addressing a critical concern in the semi-supervised deep learning modeling process when certain clients only have access to unlabeled data [13].

The existing semi-supervised federated learning methods are constrained by unreliable information hidden in the corresponding local model and struggle to optimize the performance of the federation model. Therefore, it is important to design a reliable screening mechanism for the local model and guide the federated learning process. 

This paper starts from the perspective of a reliable information-screening mechanism for the corresponding local model. Then, a dynamic semi-supervised federated learning method based on an attention mechanism is proposed, aiming to solve the problem of negative migration for federated learning due to unreliable information from a low-quality local model and improve the classification ability of clients without labeled data.

The main contributions of this work are as follows: A dynamic semi-supervised federated learning fault diagnosis method based on an attention mechanism is proposed to solve the problem of negative transfer due to unreliable information hidden in a local model. This guarantees the performance of the federation model and enhances the classification ability of clients without labeled data.A federation strategy driven by an attention mechanism is designed to filter out unreliable information so that the federation model can incorporate useful information from an unreliable local model. A new loss function related to supervised classification, unsupervised feature reconstruction, and the reliability of the local model is designed to train the federation model. According to the reliability of the federation model, the local model can be optimized by dynamically adjusting how the unlabeled data are utilized and the extent to which they can contribute.In cases where there are certain clients without labeled data, the method proposed in this study can still ensure the performance of the federation model and render it capable of fault classification for local clients without labeled data. 

## 2. Related Work

### 2.1. Semi-Supervised Deep-Learning-Based Fault Diagnosis Method

Semi-supervised deep-learning methods can achieve the full utilization of massive collections of unlabeled data to optimize fault diagnosis models built with a small quantity of labeled data [14]. The existing semi-supervised deep-learning methods can mainly be classified into generative semi-supervised methods, semi-supervised methods based on consistency regularization, graph-based semi-supervised methods, and semi-supervised methods based on pseudo-label self-training [15].

In generative semi-supervised methods, it is assumed that all samples are from the same latent model, and unlabeled data are treated as missing parameters of the potential model. An expectation maximization algorithm (EM) is usually used to determine the parameters [16,17]. Semi-supervised methods based on consistent regularization are designed to improve model robustness using unlabeled data by making predictions as consistent as possible for unlabeled data with different perturbations [18,19]. In graph-based semi-supervised learning methods, the connections among data are used to map a dataset into a graph, and then the similarity among samples is used for label propagation to achieve label prediction for unlabeled data [20,21].

Compared to the above semi-supervised deep-learning methods, the process of the semi-supervised learning method based on pseudo-label self-training is simpler and more effective [22]. Model performance can be improved via supervised learning using unlabeled data. Yu et al. [23] proposed a semi-supervised learning method that enhances the consistency of feature distribution between labeled and unlabeled data and improves the accuracy of fault diagnosis. Liu et al. [24] proposed a semi-supervised deep-learning method that alternately optimizes the pseudo-label and model parameters. The above methods use unlabeled data to improve the performance of a model through supervised learning, but the quality of the pseudo-label greatly affects the fault diagnosis performance of a model. A model’s performance can be increased by improving the quality of the pseudo-label. Ribeiro et al. [25] used a model’s prediction results to estimate reliability and then added the unlabeled data with the highest reliability to the model’s retraining process. Pedronette et al. [26] proposed a method consisting of determining a model’s reliability using marginal scores to find the most reliable pseudo-label from the unlabeled data and adding it to the labeled dataset. The above methods improve fault diagnosis performance by improving the quality of the pseudo-label, but the impact of feature accuracy on the quality of the pseudo-label is not emphasized. Zhang et al. [27] extracted data features using a variational self-encoder, which improved the fault diagnosis accuracy of the corresponding model. Tang et al. [28] used an unsupervised network to extract unlabeled data features and then fine-tuned the model jointly with the supervised network to improve semi-supervised fault diagnosis accuracy. When clients cannot achieve adequate labeling of data, it is expected that considering the utilization of labeled data from other clients to assist the client may solve the problem of difficulty in training fault diagnosis models caused by a lack of labeled data.

### 2.2. Semi-Supervised Federated Learning Fault Diagnosis Method

Semi-supervised federated learning is a method that combines semi-supervised learning and federated learning [29] to address the difficulty of achieving satisfactory fault diagnosis for clients with few labeled data and a massive number of unlabeled data. Albaseer et al. [30] proposed a semi-supervised federated learning method called FedSem, which used a federation model to assign pseudo-labels to unlabeled data and added them to the model retraining process. Diao et al. [31] proposed a semi-supervised federated learning method that is executed via alternating training by fine-tuning a federation model with labeled data and assigning pseudo-labels to unlabeled data using the federation model. However, poor quality of the pseudo-label leads to the degradation of model performance, and the problem of the pseudo-label being unreliable can be eliminated using active learning. Presotto et al. [32] combined active learning and label propagation algorithms to improve model performance by periodically using unlabeled data assigned a pseudo-label for local model training and then aggregating the models. However, the above methods only consider how to assign a high-quality pseudo-label, ignoring the fact that the potential fault feature information hidden in unlabeled data can also be used to assist in model building. Hou et al. [33] proposed a semi-supervised federated learning model called ANN-SSFL. In this approach, clients without labeled data acquired fault features through an autoencoder, and clients with labeled data trained the classifier through supervised learning. Both clients without labeled data and those with labeled data could contribute to the federation model. However, the above methods only consider how to make full use of fault feature information from unlabeled data, ignoring the model optimization effect from the information interaction occurring in federated learning. Shi et al. [34] proposed a personalized semi-supervised federated learning method called UM-pFSSL. This method allows each client to select models from other clients that contribute to the prediction of unlabeled data. The model performance of each client is improved by aggregating only the parameters of selected models. Itahara et al. [35] improved the fault diagnosis ability of a model by exchanging the model output of clients based on the idea of knowledge distillation, using it to label data in the public dataset, and then the local model was further trained using the newly labeled data. The above method improves the federation model’s performance from a feature extraction perspective. However, unreliable information hidden in clients’ data will inevitably degrade the performance of the federation model, so determining how to filter out unreliable information is an urgent problem that needs to be solved.

## 3. Dynamic Semi-Supervised Federated Learning Fault Diagnosis Method Based on an Attention Mechanism

When there are certain clients without labeled data, existing semi-supervised federated learning methods can suffer from performance degradation due to an inability to screen for unreliable information. To ensure the performance of a federation model, a federation aggregation strategy based on an information reliability screening mechanism is necessary. This paper proposes a dynamic semi-supervised federated learning method for fault diagnosis based on an attention mechanism, whose main processes include a dynamic local training mechanism based on model performance, with the aim of dynamically adjusting the way and degree to which unlabeled data are used according to the performance of the federated model. On the other hand, local model optimization is achieved through supervised and unsupervised loss. An optimal federation aggregation strategy driven by reliable information screening is designed to filter reliable information by measuring the difference between the local model and the federated model through the attention mechanism and reflecting the contribution of each client using the attention score. The aim of this process is to filter out unreliable information through the attention mechanism and thus ensure the federated model’s performance. The block diagram of the dynamic semi-supervised federated learning fault diagnosis method based on an attention mechanism is shown in Figure 1. 

### 3.1. Dynamic Local Optimization Mechanism Based on Federation Performance

In this section, a dynamic unlabeled data utilization strategy is designed to dynamically adjust the way and the extent to which unlabeled data are used based on the performance of the federation model. The specific steps are as follows.

Step 1: When the federation model is unreliable, the recursive optimization of the local model is achieved using unlabeled data.

After receiving model parameters from the federation center, they are used as the initialization parameters of the local model, as shown in Equation (1).
(1)$$\theta_{j,0}=\theta_{FL,0}$$

Client j uses local data $D_{j}=\{\{x_{l,j},y_{l,j}\},\{x_{u,j}\}\}$ for model training. The feature extraction network is trained via multi-scale recursive feature reconstruction using unlabeled data. The forward propagation and parameter update processes are shown in Equations (2) and (3), respectively:(2)$$\left\{\begin{array}{c} Feature_{x_{u,j}}=f_{en}(x_{u,j},\theta_{en,j,0})\\ {\hat{x}}_{u,j}=f_{de}(Feature_{x_{u,j}},\theta_{de,j,0}) \end{array}\right.$$
(3)$$\left\{\begin{array}{c} \theta_{en,j,1}=\theta_{en,j,0}-\eta \frac{\partial }{\partial \left(\theta_{en,j,0}\right)}Loss_{u,j,0}\\ \theta_{de,j,1}=\theta_{de,j,0}-\eta \frac{\partial }{\partial \left(\theta_{de,j,0}\right)}Loss_{u,j,0} \end{array}\right.$$
where $f_{en}(\cdot ,\cdot )$ and $f_{de}(\cdot ,\cdot )$ denote the encoding and decoding networks, respectively; η denotes the learning rate; $Loss_{u,j,0}$ denotes the multi-scale recursive feature reconstruction loss obtained using unlabeled data; $\theta_{en,j,0}$ and $\theta_{en,j,1}$ denote the parameters of the encoding network before and after updating, respectively; and $\theta_{de,j,0}$ and $\theta_{de,j,1}$ denote the parameters of the decoding network before and after updating, respectively.

Step 2: Dynamic local semi-supervised training based on the degree of pseudo-label utilization.

In the $r^{th}$ round of federated learning, the federation center distributes model parameters to the clients. Client j uses the received federation model to predict the category information of unlabeled data, as shown in Equation (4).
(4)$$y_{u,j,pre}=f_{classifier}\left(f_{en}\left(x_{u,j},\theta_{en,Fl,r}\right),\theta_{classifier,Fl,r}\right)$$

Above, $\theta_{en,Fl,r}$ denotes the encoding parameters of the $r^{th}$-round federation model, while $\theta_{classifier,Fl,r}$ denotes the classifier parameters of the $r^{th}$-round federation model. The category with the maximum probability is taken as the federation model’s pseudo-label, as shown in Equations (5) and (6), where C is the total number of categories.
(5)$${\hat{y}}_{u,j,c}=\left\{\begin{array}{c} 1\;c=\mathrm{argmax}(y_{u,j,pre})\\ 0\;c=\mathrm{argmax}(y_{u,j,pre}) \end{array}\right.$$
(6)$${\hat{y}}_{u,j}=\left[{\hat{y}}_{u,j,1},{\hat{y}}_{u,j,2},\cdots ,{\hat{y}}_{u,j,C}\right]$$

The local loss function is constructed using the classification loss of the labeled data and that of the unlabeled data with a pseudo-label, as shown in Equation (7):(7)$$Loss_{j,r}=Loss\left(x_{l,j},\theta_{j,r}\right)+\alpha (r)Loss\left(x_{u,j},\theta_{j,r}\right)$$
where α(r) is a balance parameter between the classification loss of labeled data and the classification loss of unlabeled data with a pseudo-label, which can be dynamically adjusted according to the performance of the federation model, allowing the utilization degree of the unlabeled data to be changed during the training of the local semi-supervised model. 

In this study, the number of communication rounds was used as a measure of federation model performance, and α(r) was determined according to the maximum utilization of unlabeled data $\alpha_{max}$, the current number of federation communications r, and the maximum number of federation communications R, which are calculated as shown in Equation (8).
(8)$$\alpha (r)=\alpha_{max}\frac{r}{R}$$

In this step, the local model parameters are updated, as shown in Equation (9).
(9)$$\theta_{j,r+1}=\theta_{j,r}-\eta \left(\frac{\partial }{\partial \left(\theta_{j,r}\right)}Loss\left(x_{l,j},\theta_{j,r}\right)+\alpha \left(r\right)\frac{\partial }{\partial \left(\theta_{j,r}\right)}Loss\left(x_{u,j},\theta_{j,r}\right)\right)$$

The above steps enable the utilization of labeled and unlabeled data in a supervised learning manner, and the differential utilization of unlabeled data can achieve the goal of the full use of large collections of unlabeled data from each client for model optimization.

### 3.2. Federation Strategy Driven by Screening of Reliable Information 

Unreliable information is hidden in the local model because it is difficult to achieve high-quality model building for clients without labeled data. Therefore, at this stage, a semi-supervised federation aggregation strategy based on an attention mechanism is designed. The specific steps are as follows.

Step 1: Semi-supervised federation model aggregation.

After receiving the local model uploaded by all clients, the federation center aggregates all local models using the initialized federation aggregation parameters, as shown in Equation (10).
(10)$$\theta_{FL}=\sum_{j=1}^{J}Pk_{j}⊙\theta_{j}$$

Above, $\theta_{FL}$ is the federation model, and $Pk_{j}$ is the aggregation weight of Client j.

Step 2: Establish model reliability metrics based on the degree of consistency.

The local model parameters are used as query values to query the attention distribution between each local model and federation model in turn, and the attention score is scaled to between 0 and 1. The contribution degree of the local model $\theta_{j}$ is calculated as shown in Equation (11):(11)$$Att_{j}=\frac{s\left(\theta_{j},\theta_{FL}\right)}{∥\theta_{j}∥\times ∥\theta_{FL}∥}$$
where s(·,·) is the attentional evaluation function, which is the dot product, as shown in Equation (12):(12)$$s\left(q,k\right)=q^{T}k$$
where q is the query vector in the attention mechanism, k is the key vector in the attention mechanism, and s(q,k) represents the attention score between q and k.

Step 3: The federation aggregation process is driven by the performance of the federation model and the reliability of the local model.

The loss function of the dynamic semi-supervised federation aggregation process is designed by integrating federation model performance and the local model reliability of the clients, as shown in Equation (13):(13)$$Loss_{FL}=\sum_{j=1}^{J}Loss_{x_{l},j}+\sum_{j=1}^{J}Loss_{x_{u},j}+\sum_{j=1}^{J}\frac{1}{Att_{j}}\;+L_{2,Pk}$$
where $Loss_{x_{l},j}$ denotes the supervised loss for Client j, as shown in Equation (14); $Loss_{x_{u},j}$ denotes the unsupervised loss for Client j, as shown in Equation (15); and J denotes the total number of clients. $L_{2,Pk}$ is the regularization term that restricts the sum of the federation aggregation weights to 1, as shown in Equation (16):(14)$$Loss_{x_{l},j}=-\frac{1}{M_{l,j}}\sum_{m=1}^{M_{l,j}}y_{l,m}\log(f_{DNN,FL}(x_{l,m},\theta_{FL}))$$
(15)$$Loss_{x_{u},j}=\frac{1}{M_{u,j}}\sum_{m=1}^{M_{u,j}}(f_{de,FL}(f_{en,FL}(x_{u,m},\theta_{en,FL}),\theta_{de,FL})-x_{u,m}{)}^{2}$$
(16)$$L_{2,Pk}=∥1-\sum_{j=1}^{J}Pk_{j}{∥}^{2}$$
where $M_{l,j}$ and $M_{u,j}$ denote the volume of labeled data and the volume of unlabeled data for Client j, respectively; $x_{l,m}$ is the $m^{th}$ labeled sample for Client j; $y_{l,m}$ is the label corresponding to the $m^{th}$ sample for Client j; $x_{u,m}$ is the $m^{th}$ unlabeled sample for Client j; $f_{DNN,FL}(\cdot ,\cdot )$ denotes the DNN model obtained via aggregation; $f_{en,FL}(\cdot ,\cdot )$ and $f_{de,FL}(\cdot ,\cdot )$ denote the encoding and decoding networks of the federation model, respectively; and the corresponding network parameters are denoted by $\theta_{en,FL}$ and $\theta_{en,FL}$, respectively.

Step 4: Joint optimization of local model parameters and federation aggregation weights.

The joint optimization of local model parameters and federation aggregation weights based on the loss function of the federation center can further improve the fault diagnosis performance of the federation model by improving the reliability of the local model. The gradients of the local model parameters and the federation aggregation weights are calculated as shown in Equations (17) and (18), respectively.
(17)$$\nabla Pk_{j}=\frac{\partial }{\partial Pk_{j}}Loss_{x_{l},j}+\frac{\partial }{\partial Pk_{j}}Loss_{x_{u},j}+\frac{\partial }{\partial Pk_{j}}\frac{1}{Att_{j}}+\frac{\partial }{\partial Pk_{j}}L_{2,Pk}$$
(18)$$\nabla \theta_{j}=\frac{\partial }{\partial \theta_{j}}Loss_{x_{l},j}+\frac{\partial }{\partial \theta_{j}}Loss_{x_{u},j}+\frac{\partial }{\partial \theta_{j}}\frac{1}{Att_{j}}+\frac{\partial }{\partial Pk_{j}}L_{2,Pk}$$

$Pk_{j},\theta_{j}(j=1,2,\cdots ,J)$ can be optimized according to the obtained gradient. Thus, the federation model performance and local model reliability can be used to jointly drive the federation aggregation process. The parameter-updating process is shown in Equations (19) and (20):(19)$$Pk_{j,r+1}=Pk_{j,r}-\eta \nabla Pk_{j}$$
(20)$$\theta_{j,r+1}=\theta_{j,r}-\eta \nabla \theta_{j}$$
where $Pk_{j,r}$ and $\theta_{j,r}$ denote the aggregation weights and local model parameters of Client j in round r, while $Pk_{j,r+1}$ and $\theta_{j,r+1}$ denote the aggregation weights and local model parameters of Client j after updating.

The joint optimization process can dynamically update the federation aggregation weights and local model parameters, which improves the reliability of local models and thus reduces the impact of unreliable local models on the performance of the federation model. 

### 3.3. Fault Diagnosis Based on SSFL-ATT

This section highlights the detailed steps of the SSFL-ATT fault diagnosis method proposed to solve the problem of reliable information screening for local models.

A flowchart of the dynamic semi-supervised federated learning algorithm based on the attention mechanism is shown in Figure 2. Fault diagnosis is divided into offline training and online diagnosis. In the offline training part, the green box is a dynamic local training mechanism based on model performance, and the blue box is the optimal federation aggregation strategy driven by both model performance and consistency metrics. In the fault diagnosis part, clients feed the preprocessed data into the federation model to obtain fault features, and the fault features are fed into the classifier to obtain the diagnosis results. The detailed steps are formally represented in Algorithm 1.
**Algorithm 1:** Fault diagnosis based on SSFL-ATTRequire: local data XkServer executes: Initializing federation model θFL,0Step1: Model training for semi-supervised federated learningDynamic local training mechanism based on federation model performanceClients dynamically adjust how to use local unlabeled data based on federation model performance.Lossj,r=Lossxl,j,θj,r+αrLossxu,j,θj,rθj,r+1=θj,r−η∂∂θj,rLossj,rFederation aggregation strategy driven by reliable information screeningReliable information can be screened from local models based on attention mechanismsAttj=sθj,θFL∥θj∥×∥θFL∥The loss function can be designed by combining performance of federation model and reliability of local modelLossFL=∑j=1JLossxl,j+∑j=1JLossxu,j+∑j=1J1Attj +L2,PkJoint optimization of local model parameters and federation aggregation weights.Joint optimization of local model parameters and coalition aggregation weights can be achieved based on loss function of federation centerPkj,r+1=Pkj,r−η∂∂Pkj,rLossFL*,*
θj,r+1=θj,r−η∂∂θj,rLossFLθFL=∑j=1JPkj⊙θjStep2: Fault diagnosis for each clientEach client uses well-trained federation model θFL to achieve fault diagnosis

## 4. Experiment and Analysis

### 4.1. Experimental Analysis of the Bearing Fault Simulation Platform at Case Western Reserve University

Using the benchmark dataset of Case Western Reserve University (CWRU) for experimental verification, this section validates the effectiveness of the proposed method through specific experimental analysis.

#### 4.1.1. Bearing Data Description

The Case Western Reserve University bearing dataset is widely used in fault diagnosis. The experimental platform is shown in Figure 3 [36]. It mainly consists of a three-phase asynchronous motor, a torque transducer or decoder, and a power test meter. A single-point fault was introduced in the motor bearing using electro-discharge-machining (EDM) techniques with fault sizes of 0.007 in, 0.014 in, and 0.021 in. Accelerometers were installed at the drive and fan ends to collect vibration data for motor loads of 0 HP to 3 HP.

In this section of the experiment, the monitoring signal of the accelerometer at the drive end was selected. The motor load was 0 HP, the speed was 1797 rpm, and the sampling frequency was 48 KHz. The four operation states of the rolling bearing comprise the normal operation state and the fault states of the inner ring, ball, and outer ring measured at a fault inch of 0.021. The detailed composition of the dataset is shown in Table 1.

To verify the superiority of the proposed method, it was compared with various existing models of semi-supervised federated learning. Table 2 describes the different models established in this section during the experimental validation and the experimental parameter settings.

An experimental scenario of semi-supervised federated learning was designed by changing the data of the clients. The dataset was obtained by intercepting the vibration data through a sliding window with a window size of 400 and a step size of 30, and the number of samples in the test set is 4 × 300. Table 3 shows the design of the experiment.

#### 4.1.2. Bearing Experiment Results and Analysis

Experiments 1–3 are designed to verify the effectiveness and superiority of SSFL-ATT when clients have different quantities of labeled data. The fault diagnosis results are shown in Table 4.

As can be seen in Columns 3 and 4 of Table 4, the use of only unsupervised and supervised learning methods resulted in some clients failing to achieve fault diagnosis. As shown in Columns 4 and 5, supervised learning was used to train local models. Then, the federated averaging algorithm was used to aggregate the models. Finally, the federation model was distributed to all the clients. Client 3 realized fault diagnosis without labeled data, but the diagnostic accuracy was poor. This was because the data for Client 3 were not involved in the model training to extract the corresponding fault information. As gleaned when comparing Columns 5 and 6, FedSem used the federation model to assign pseudo-labels to the unlabeled data of all the clients, in which the fault information hidden in the unlabeled data of Client 3 was fully utilized. However, negative transfer occurred in the federation model due to the unreliability of the pseudo-label. This led to the aggregation of unreliable information during the federation process. When comparing Columns 6 and 7, it is clear that Sem-Fed utilized the local model to assign a pseudo-label to unlabeled data, and therefore, ensured that the pseudo-label was not affected by unreliable information from other clients, enhancing the reliability of the pseudo-label and thus improving the performance of the federated model. As observed upon comparing Columns 7 and 8, different from FedSem, ANN-SSFL constructed the loss function of the federation center according to the accuracy of the feature representation for the unlabeled data, causing the unlabeled data to contribute sufficiently to the federation center, thus guaranteeing the comprehensiveness of the information utilization. However, unreliable information in unlabeled data were also aggregated in the federation model. Upon comparing Columns 8 and 9, it is clear that SSFL-ATT obtained better fault diagnosis accuracy than ANN-SSFL. SSFL-ATT filters out unreliable model information from the local model through the attention mechanism. And the loss function of the federation center can be used to guide the joint optimization of federation aggregation weights and local model parameters, thus guaranteeing the reliability of the federation model through the precise utilization of reliable information.

To further validate the effectiveness of the proposed method, the confusion matrices of each fault diagnosis method shown in Experiment 2 are given in Figure 4, Figure 5, Figure 6, Figure 7 and Figure 8.

Upon comparing the confusion matrices of Figure 6, Figure 7 and Figure 8, it is clear that the diagnostic accuracy of both the health data and outer-race fault data was significantly degraded. This shows that unreliable information caused by local pseudo-label led to negative transfer in the federation model. As gleaned when comparing Figure 4, Figure 5, Figure 6, Figure 7, Figure 8 and Figure 9 with Figure 10, the diagnostic accuracy was close to 100% for both the health data and outer-race fault data, and the similar features of the inner-race fault data and ball fault data are easier to recognize. This finding shows that the proposed method can prevent unreliable model information from interfering with the federation model’s performance and improve the performance of the federation model through an effective federation aggregation strategy.

This section analyzes the evolution of the aggregation weights, *Pk*, and attention scores, *Att*, in Experiment 2. As shown in Figure 11, we plotted the attention scores of the clients at different stages on a graph to see how the attention scores changed. In the beginning, Client 1 and Client 2 had high attention scores, while Client 3 had a low attention score. This is because the performance of the federation model in the early stages was not high enough to assign a high-quality pseudo-label to unlabeled data. This resulted in a more erroneous pseudo-label for Client 3, which had a poorer-quality local model, and hence, a low attention score. As the federation proceeded, the attention scores of all the clients rose, and the attention score of Client 3 rose rapidly. This is because SSFL-ATT screened out unreliable information through an attention mechanism to ensure the performance of the federation model, which, in turn, ensured the quality of the pseudo-label for unlabeled data. Therefore, the quality of the local model was ensured, and, eventually, higher attention scores were obtained. In the late stage, the attention scores of the clients were similar because they all had high-quality pseudo-labels assigned by the federation model, leading to an improvement in the performance of the local model. It is worth noting that Client 3 had a higher attention score than the other clients. This is because Client 3 had a massive number of samples to which the federation model gave a high-quality pseudo-label, and Client 3, therefore, attained a high-quality local model, which, in turn, gave the model a higher attention score. Since *Pk* is a matrix, we determined the mean value of *Pk* and then plotted its evolutionary trend; as shown in Figure 12, the evolutionary trend of *Pk* is like that of *Att*, which further shows that the method proposed in this paper can filter reliable information and improve the performance of the federation model.

Experiments 4–6 were designed to verify the improvement induced by SSFL-ATT on the performance of the fault diagnosis model when clients have different quantities of unlabeled data. The experimental results are shown in Table 5.

Compare Columns 3–8 and 9 of Experiment 3 and Experiments 4–6. This comparison reveals that when the quantity of the labeled data is fixed, the fault diagnosis accuracy of all the methods decreases as the quantity of unlabeled data decreases. However, SSFL-ATT still achieved high classification accuracy, which indicates that SSFL-ATT was better able to extract reliable information from unreliable local models for clients without labeled data, thus achieving the aim of reducing the perturbation caused by unreliable pseudo-label information sent to the local model.

### 4.2. Experimental Analysis of Motor Fault Simulation Platform at Shanghai Maritime University

Using the Shanghai Maritime University motor dataset for experimental validation, this section verifies the effectiveness of the proposed method through specific experimental analysis.

#### 4.2.1. Motor Dataset Description

The Shanghai Maritime University motor fault simulation experiment bench is shown in Figure 13. This bench consists of a drive motor, a magnetic powder brake set, a tachometer, a torque sensor, several single-axis acceleration sensors, several current clamps, and an eight-channel portable data acquisition system. Ten different datapoints of motor operation states were used in this section. The data were collected using a drive-end vibration sensor, a fan-end vibration sensor, three current clamps, a torque sensor, and a speed sensor at a sampling frequency of 12,800 Hz. The detailed composition of the dataset is provided in Table 6.

During the experimental validation, DNN was used as the base network model, with the number of neurons in each layer set to 7/200/90/30/10 and a learning rate of 0.005 applied, and model optimization was performed using the Adam optimizer. The number of samples in the test set was 10 × 1000. The detailed design of the experiment is in Table 7.

#### 4.2.2. Motor Experimental Results and Analysis

In order to verify the effectiveness of SSFL-ATT with multi-channel signals, the results of Experiments 1–3 are shown in Table 8.

As can be seen in Table 8, the SSFL-ATT method was still superior to the other methods in the multi-channel signal experiments. Although the multi-channel signals contained richer information than the single-channel signals, the amount of unreliable information in the multi-channel signals also increased exponentially. Therefore, a reliable-information-screening mechanism was needed to filter out the unreliable information contained in the multichannel data, thus ensuring the adequate performance of the federal model. SSFL-ATT uses an attention mechanism to filter out unreliable information hidden in local models and enhance clients’ classification ability without labeled data. In the case of multi-channel data, Experiment 4 and Experiment 5 were designed to verify the fault diagnosis effect of the proposed method when the clients have different quantities of unlabeled data. The experimental results are shown in Table 9.

SSFL-ATT was still superior to the other methods in the above two experimental scenarios. From the experimental results in Table 9, it can be concluded that SSFL-ATT is still superior to the other methods, which further indicates that SSFL-ATT can still achieve reliable information screening and be used to build a well-performing federated model under multi-channel data. The experimental results from the Case Western Reserve University’s benchmark dataset and the motor fault dataset of Shanghai Maritime University show that SSFL-ATT was applicable and superior in relation to both the single-channel and multi-channel signals for fault diagnosis.

To illustrate the superiority of SSFL-ATT more intuitively in different experimental scenarios with two datasets, Figure 10 shows the histograms of all the experimental results. From Figure 14, it can be gleaned that SSFL-ATT was substantially improved in different experimental scenarios with both single-channel and multi-channel data. This result indicates that SSFL-ATT is more applicable and superior compared to other methods in terms of semi-supervised federated learning fault diagnosis.

## 5. Conclusions

In semi-supervised federated learning for fault diagnosis, when there are certain clients without labeled data, existing semi-supervised federated learning methods can lead to a negative transfer problem due to unreliable information hidden in these clients’ local models. This paper proposes a dynamic semi-supervised federated learning fault diagnosis method based on an attention mechanism for designing an optimal federation aggregation strategy. The federation aggregation strategy was dynamically optimized based on reliable information screened in the local model.

First, to ensure the effectiveness of its utilization for local unlabeled data, clients can dynamically adjust the way and extent to which unlabeled data are used according to the performance of the federation model. The aim is to fully utilize unlabeled data for model optimization while reducing the perturbation of the local training process with low-quality pseudo-labels. Then, in the process of federation aggregation, the occurrence of negative transfer in federated learning due to unreliable model information can be avoided by establishing reliability evaluation metrics based on the attention mechanism. At the same time, the feedback information from clients on the performance of the federation model can be combined to drive the federation aggregation process and achieve the joint optimization of aggregation weights and local model parameters. The experimental results show that SSFL-ATT can utilize an attention mechanism to filter out unreliable information to avoid negative transfer caused by unreliable information, and it can also effectively improve the classification ability of unlabeled clients. Compared to existing semi-supervised federal learning methods (a comparison shown in in the experiments of both single-channel and multi-channel signals), these results indicate that SSFL-ATT is superior and more applicable.

## Figures and Tables

**Figure 1 entropy-25-01470-f001:**
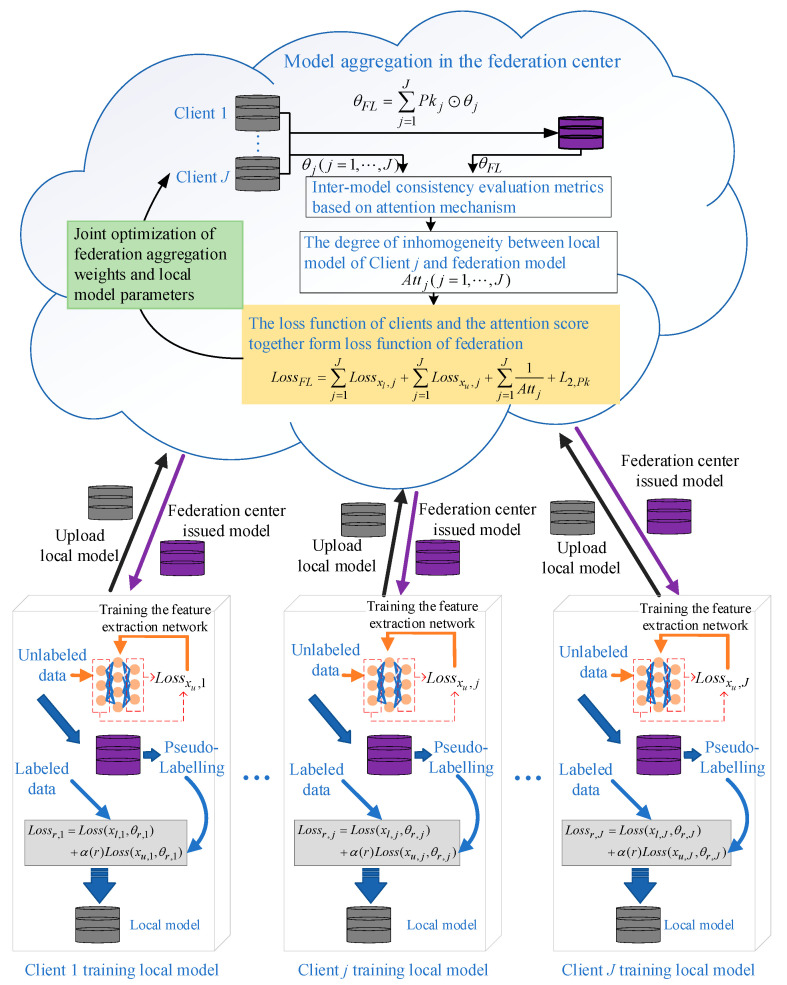
The algorithm of the dynamic semi-supervised federated learning fault diagnosis method based on an attention mechanism.

**Figure 2 entropy-25-01470-f002:**
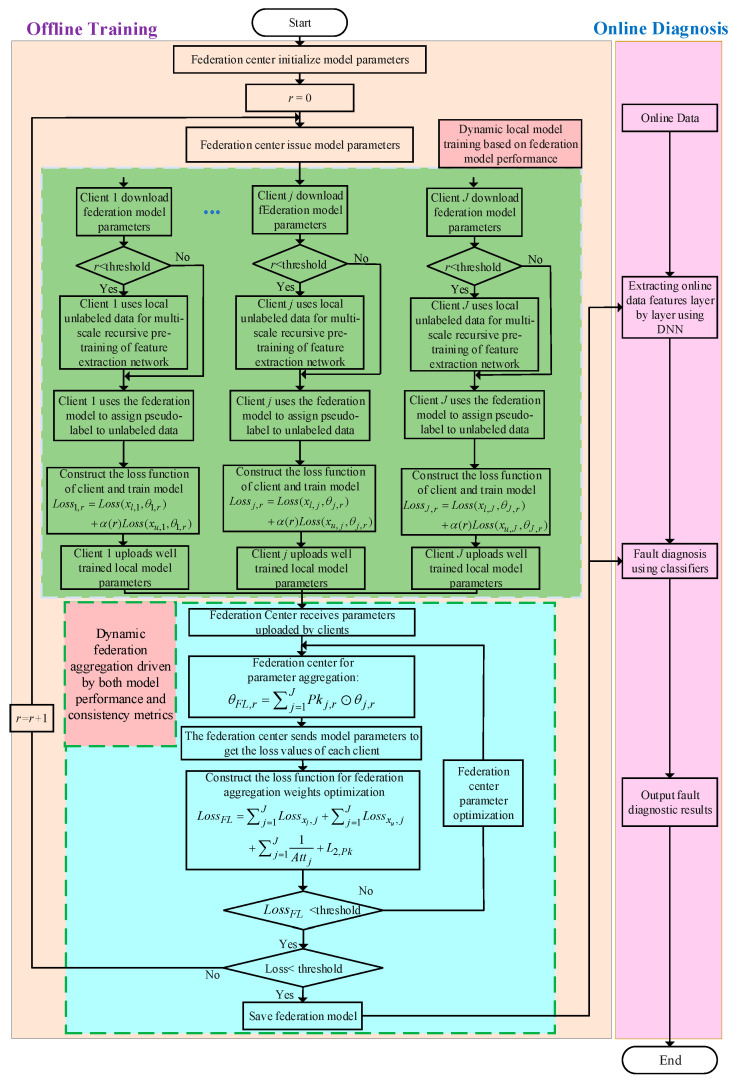
Flowchart of the dynamic semi-supervised federated learning fault diagnosis method based on an attention mechanism.

**Figure 3 entropy-25-01470-f003:**
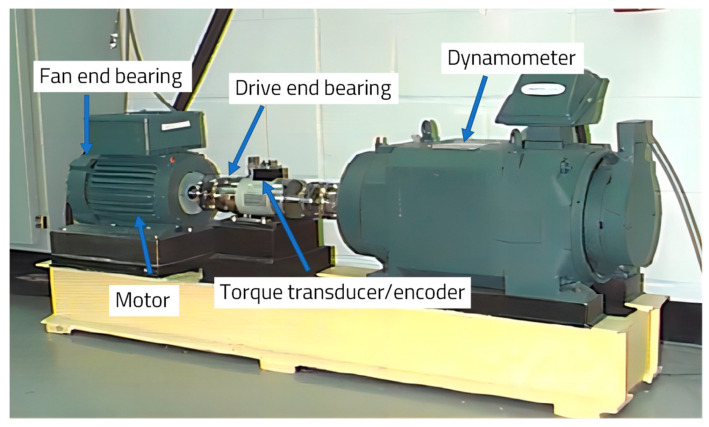
The Case Western Reserve University bearing experiment bench [36].

**Figure 4 entropy-25-01470-f004:**
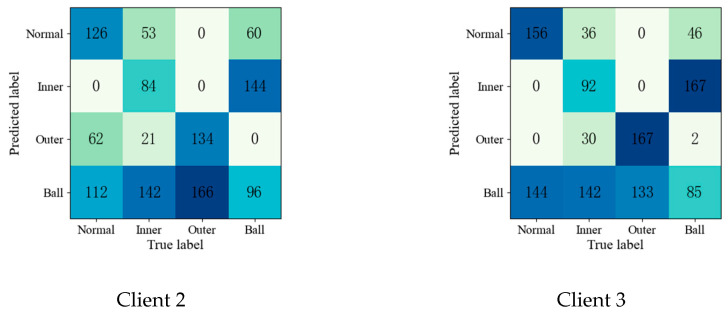
Confusion matrix of feature clustering for fault diagnosis result.

**Figure 5 entropy-25-01470-f005:**
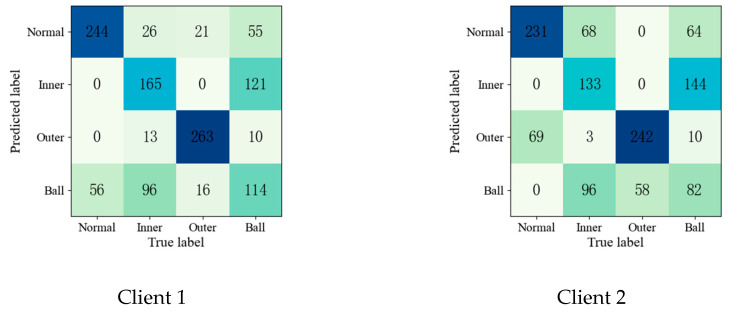
Confusion matrix of DNN for fault diagnosis result.

**Figure 6 entropy-25-01470-f006:**
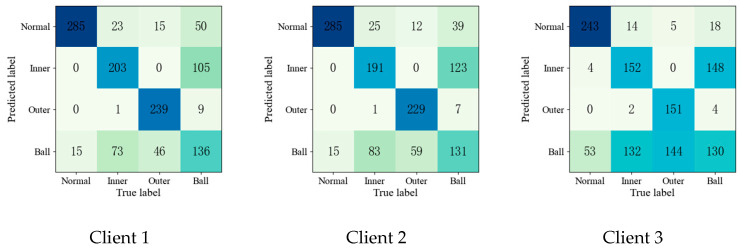
Confusion matrix of FedAvg for fault diagnosis result.

**Figure 7 entropy-25-01470-f007:**
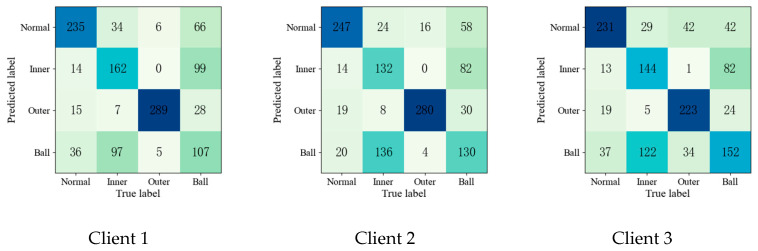
Confusion matrix of FedSem for fault diagnosis result.

**Figure 8 entropy-25-01470-f008:**
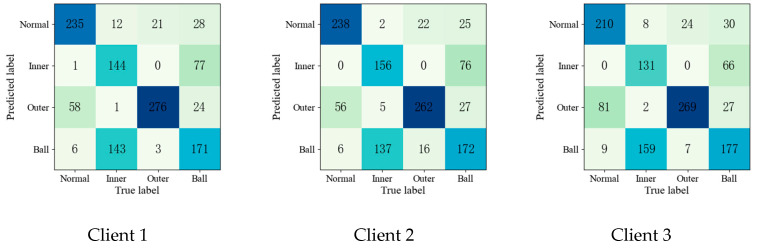
Confusion matrix of Sem-Fed for fault diagnosis result.

**Figure 9 entropy-25-01470-f009:**
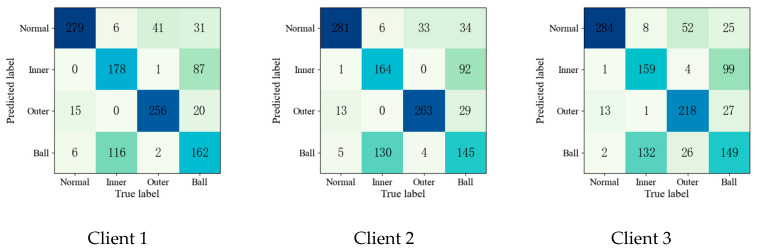
Confusion matrix of ANN-SSFL for the fault diagnosis result.

**Figure 10 entropy-25-01470-f010:**
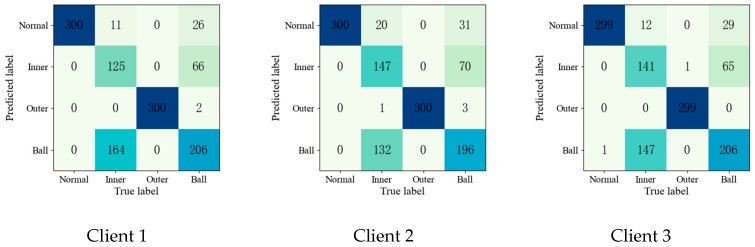
Confusion matrix of SSFL-ATT for the fault diagnosis result.

**Figure 11 entropy-25-01470-f011:**
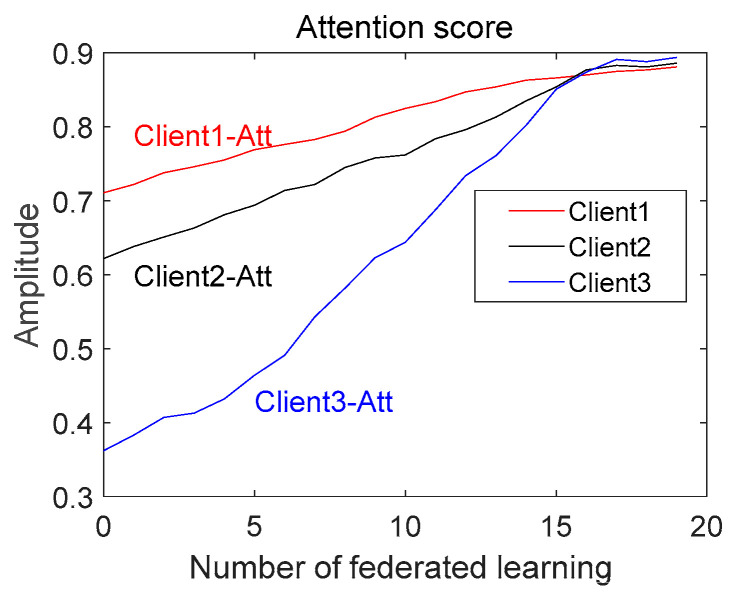
Evolution of attention scores (Att).

**Figure 12 entropy-25-01470-f012:**
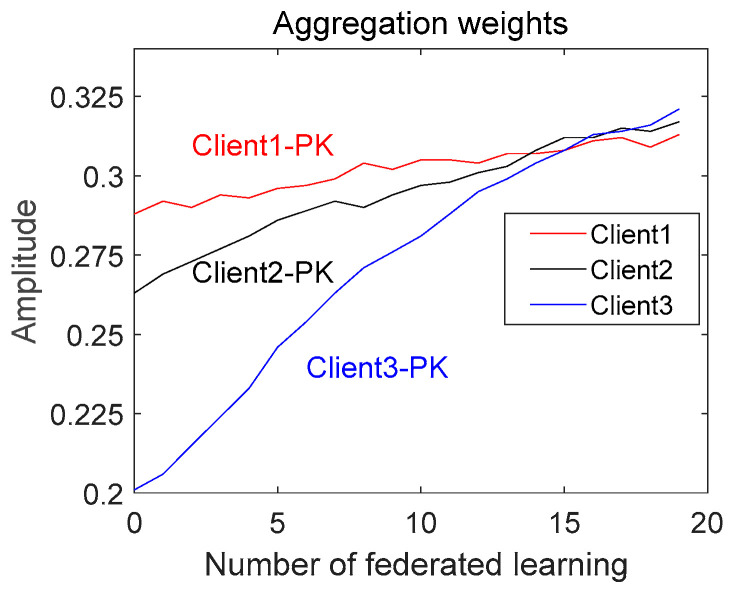
Evolution of aggregation weights (Pk).

**Figure 13 entropy-25-01470-f013:**
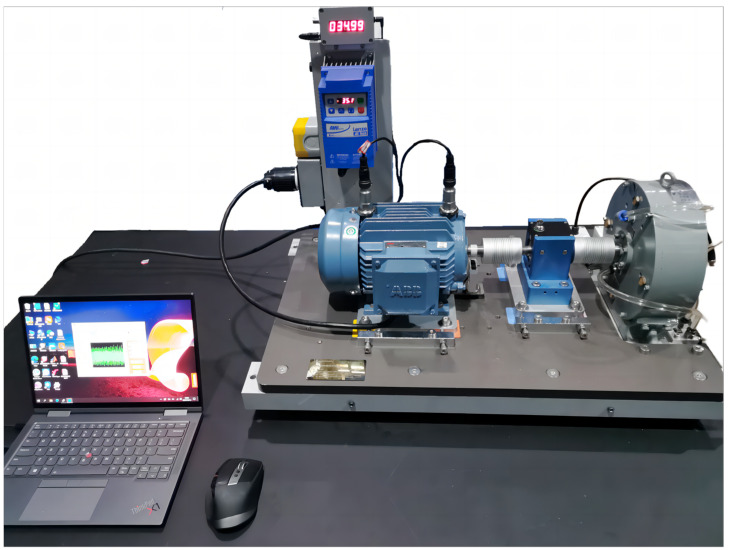
Shanghai Maritime University motor fault simulation experiment bench.

**Figure 14 entropy-25-01470-f014:**
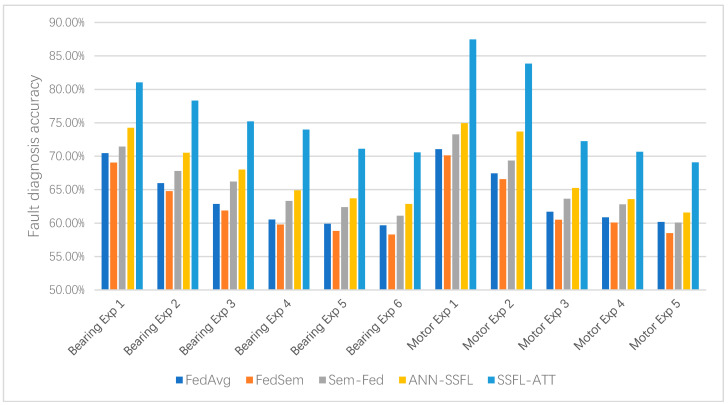
Histogram of the experimental results of different semi-supervised federated learning fault diagnosis methods.

**Table 1 entropy-25-01470-t001:** Bearing data types.

Fault Type	Failure Size (Inch)	Label
Normal data	0	Normal
Inner ring failure	0.021	Inner
Outer ring failure	0.021	Outer
Ball failure	0.021	Ball

**Table 2 entropy-25-01470-t002:** Semi-supervised federated learning model and parameter settings.

Model	Brief Description of the Model	Model Parameters	Number of Clients
Feature-Clustering	Fault diagnosis is achieved via clustering after extracting fault features; then, unsupervised learning is used to execute fault diagnosis.	Clustering centers: 4	1
DNN	Traditional deep learning fault diagnosis model.	Number of network layers: 5 Number of neurons in each layer: 400/500/100/30/4 Learning rate: 0.005	1
FedAvg [29]	The local model is trained using supervised learning. The federated averaging algorithm is used to obtain a federation model, which is downloaded by all clients to achieve fault diagnosis.		3
FedSem [30]	The federation model assigns a pseudo-label to unlabeled data and then adds them to the model-retraining process.		3
Sem-Fed	The local model is used to assign a pseudo-label to unlabeled data, and then the federated averaging strategy is used to aggregate models.		3
ANN-SSFL [33]	The local model is jointly trained by determining unsupervised and supervised loss and then aggregated via the federated averaging algorithm.		3
SSFL-ATT	Dynamic semi-supervised federated learning fault diagnosis method based on attention mechanism.		3

**Table 3 entropy-25-01470-t003:** Experimental design.

Experiment	Quantity of Data for Client 1		Quantity of Data for Client 2		Quantity of Data for Client 3	
	Labeled	Unlabeled	Labeled	Unlabeled	Labeled	Unlabeled
Experiment 1	4 × 100	0	4 × 50	4 × 1000	0	4 × 5000
Experiment 2	4 × 50	0	4 × 20	4 × 1000	0	4 × 5000
Experiment 3	4 × 20	0	4 × 10	4 × 1000	0	4 × 5000
Experiment 4	4 × 20	0	4 × 10	4 × 1000	0	4 × 3000
Experiment 5	4 × 20	0	4 × 10	4 × 1000	0	4 × 2000
Experiment 6	4 × 20	0	4 × 10	4 × 1000	0	4 × 1500

**Table 4 entropy-25-01470-t004:** Bearing fault diagnosis results for different quantities of labeled data.

Experiment	Client	Feature Clustering	DNN	FedAvg	FedSem	Sem-Fed	ANN-SSFL	SSFL-ATT
Experiment 1	Client 1	—	71.50%	75.50%	70.67%	72.67%	76.83%	82.08%
	Client 2	37.17%	64.50%	74.08%	69.17%	73.41%	75.00%	81.50%
	Client 3	44.25%	—	61.75%	67.33%	68.17%	70.92%	79.50%
	Mean	—	—	70.44%	69.06%	71.42%	74.25%	81.03%
Experiment 2	Client 1	—	65.50%	71.92%	66.08%	68.83%	72.92%	77.58%
	Client 2	36.67%	57.33%	69.67%	65.75%	69.00%	71.08%	78.58%
	Client 3	44.17%	—	56.33%	62.50%	65.58%	67.50%	78.75%
	Mean	—	—	65.97%	64.78%	67.80%	70.50%	78.30%
Experiment 3	Client 1	—	56.92%	67.83%	63.41%	67.58%	68.25%	74.83%
	Client 2	35.41%	51.67%	66.33%	62.17%	66.08%	68.83%	75.17%
	Client 3	44.92%	—	54.41%	60.08%	65.00%	66.92%	75.58%
	Mean	—	—	62.86%	61.89%	66.22%	68.00%	75.19%

**Table 5 entropy-25-01470-t005:** Bearing fault diagnosis results for different quantities of unlabeled client data.

Experiment	Client	Feature Clustering	DNN	FedAvg	FedSem	Sem-Fed	ANN-SSFL	SSFL-ATT
Experiment 4	Client 1	—	56.58%	63.41%	60.58%	64.17%	65.58%	73.58%
	Client 2	37.50%	52.08%	63.33%	59.92%	63.41%	65.41%	74.83%
	Client 3	44.50%	—	54.83%	58.83%	62.33%	63.75%	73.50%
	Mean	—	—	60.52%	59.78%	63.30%	64.91%	73.97%
Experiment 5	Client 1	—	56.25%	62.50%	58.50%	63.50%	64.75%	71.83%
	Client 2	36.25%	52.50%	62.83%	59.58%	62.33%	64.08%	71.33%
	Client 3	40.50%	—	54.41%	58.41%	61.33%	62.25%	70.16%
	Mean	—	—	59.91%	58.83%	62.39%	63.69%	71.11%
Experiment 6	Client 1	—	57.17%	61.92%	59.16%	61.33%	64.58%	72.25%
	Client 2	37.17%	51.17%	62.25%	58.33%	61.41%	63.00%	68.58%
	Client 3	38.83%	—	54.75%	57.33%	60.58%	61.00%	70.83%
	Mean	—	—	59.64%	58.27%	61.11%	62.86%	70.55%

**Table 6 entropy-25-01470-t006:** Motor dataset composition.

Fault Type	Fault Simulation Method	Fault Level	Label
Normal	\	0	0
Bearing inner-ring fault	Fault grooves are machined through the inner raceway of the bearing via laser etching.	0.5 mm	1
Bearing outer-ring fault	Fault grooves are machined through the outer raceway of the bearing via laser etching.	0.5 mm	2
Shaft bending fault	Pressure is applied to the rotor using a press to obtain different degrees of bending.	0.3 mm	3
Broken rotor bar fault	The milling process breaks part of the copper bar in the rotor.	Break two bars	4
Rotor imbalance fault	The local mass of the rotor is removed.	4 g	5
Misalignment fault	The bearing mounting position is widened, and bolts are used to adjust the bearing position.	0.25 mm	6
Voltage unbalance	An external control box is used to adjust the resistance value to produce different levels of voltage imbalance.	50%	7
Out-of-phase fault	External control box: the phase loss button is turned on and off.	Out of V-phase	8
Winding short-circuit fault	A short-circuit terminal is preset in the control box; the resistance value is adjusted to introduce different degrees of a winding short-circuit fault.	10%	9

**Table 7 entropy-25-01470-t007:** Design of the experiment.

Experiment	Quantity of Data for Client 1		Quantity of Data for Client 2		Quantity of Data for Client 3	
	Labeled	Unlabeled	Labeled	Unlabeled	Labeled	Unlabeled
Experiment 1	10 × 100	0	10 × 50	10 × 5000	0	10 × 10,000
Experiment 2	10 × 50	0	10 × 30	10 × 5000	0	10 × 10,000
Experiment 3	10 × 20	0	10 × 10	10 × 5000	0	10 × 10,000
Experiment 4	10 × 20	0	10 × 10	10 × 1000	0	10 × 3000
Experiment 5	10 × 20	0	10 × 10	10 × 100	0	10 × 300

**Table 8 entropy-25-01470-t008:** Motor fault diagnosis results with different quantities of label data for clients.

Experiment	Client	Feature Clustering	DNN	FedAvg	FedSem	Sem-Fed	ANN-SSFL	SSFL-ATT
Experiment 1	Client 1	—	70.45%	74.05%	71.97%	73.58%	75.32%	87.30%
	Client 2	41.50%	66.26%	74.65%	70.36%	73.66%	75.26%	87.77%
	Client 3	48.63%	—	64.45%	68.06%	72.57%	74.23%	87.33%
	Mean	—	—	71.05%	70.13%	73.27%	74.94%	87.47%
Experiment 2	Client 1	—	65.35%	71.32%	67.93%	70.96%	74.03%	83.77%
	Client 2	42.12%	61.47%	70.65%	67.04%	69.06%	73.73%	84.22%
	Client 3	47.55%	—	60.37%	64.78%	68.00%	73.25%	83.55%
	Mean	—	—	67.45%	66.58%	69.34%	73.67%	83.85%
Experiment 3	Client 1	—	59.32%	65.35%	62.05%	64.70%	66.05%	72.91%
	Client 2	41.12%	57.15%	64.35%	60.98%	63.23%	65.67%	72.49%
	Client 3	46.63%	—	55.35%	58.45%	62.98%	64.03%	71.34%
	Mean	—	—	61.68%	60.49%	63.64%	65.25%	72.25%

**Table 9 entropy-25-01470-t009:** Motor fault diagnosis results with different quantities of label data for clients.

Experiment	Client	Feature Clustering	DNN	FedAvg	FedSem	Sem-Fed	ANN-SSFL	SSFL-ATT
Experiment 4	Client 1	—	58.82%	64.35%	61.21%	63.31%	64.80%	71.79%
	Client 2	32.50%	56.84%	63.35%	60.54%	62.34%	63.69%	70.60%
	Client 3	37.45%	—	54.82%	58.45%	62.77%	62.27%	69.56%
	Mean	—	—	60.84%	60.07%	62.81%	63.59%	70.65%
Experiment 5	Client 1	—	59.32%	63.35%	60.58%	61.35%	62.62%	69.70%
	Client 2	29.35%	57.15%	62.35%	57.56%	60.20%	61.44%	69.52%
	Client 3	38.63%	—	54.82%	57.31%	58.64%	60.64%	68.06%
	Mean	—	—	60.17%	58.48%	60.06%	61.57%	69.09%

## Data Availability

The data involved in this article are presented in the article.

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
