# Peer review of "Dynamic Semi-Supervised Federated Learning Fault Diagnosis Method Based on an Attention Mechanism"

_entropy, 2023, doi:10.3390/e25101470_

Round 1
Reviewer 1 Report
The theme of the paper is of great importance for the industrial sector. The experimental part of the paper is well described and the results are clear and evident.
However, I suggest that the authors do a major revision as I encountered several critical points in reading the paper.
1) The paper is too long, the first part is an overview article, while the second part is a research article. I would propose to the authors to divide the paper into 2 articles.
2) References are plentiful. In my opinion, it is necessary to filter the most significant references and discard some of them, above all among the references [1-6] and [7-10].
3) Many concepts are repeated too many times, such as the first sentence of section 3: "In the case..... lead to negative transferring of federated learning due to unreliable information hidden in local model".
4) The purpose of the article should be found in the introduction, instead, I find it in section 3: "This paper proposes a dynamic semi-supervised federated learning method ...... reliable information."
5) The schematic in Figure 1 should be better described when cited.
6) In the section describing the Shanghai Maritime University Motor Fault Simulation Experiment Bench, enter the data of the instrumentation used, engine, brake, torque and vibration sensors, etc.
7) The site of reference 47 is not available.
Thanks to the Authors
Author Response
Ref: entropy-2533049
Title: Dynamic semi-supervised federated learning fault diagnosis method based on attention mechanism
Authors: Shun Liu, Funa Zhou *, Shanjie Tang*, Xiong Hu, Chaoge Wang and Tianzhen Wang
Dear Editors-in-Chief and Reviewers,
We deeply appreciate the time and effort you have spent in reviewing our manuscript. Thank you very much for giving us a chance to respond to the Reviewers’ comments and we would like to thank the Reviewers for their constructive comments on our manuscript. We have considered the comments of each Reviewer and made some changes and corrections in the manuscript accordingly. Our changes and response are presented in the document of Detailed entropy to Reviewers. The point-to-point answers and explanations for all comments were listed following this letter, and the modified words and sentences are marked with blue. We hope that the Editors and Reviewers will be satisfied with the revisions for the original manuscript. If you have any question about this paper, please contact us without hesitate.
Thanks and Best regards!
Yours Sincerely,
Shun Liu, Funa Zhou *, Shanjie Tang*, Xiong Hu, Chaoge Wang and Tianzhen Wang
Corresponding author: Funa Zhou, Shanjie Tang
Department of Electrical Engineering
Shanghai Maritime University
No. 1550, Linggang avenue, Pudong Dsitrict, Shanghai
E-mail: [email protected]

Reviewer 2 Report
Review: Dynamic semi-supervised federated learning fault diagnosis method based on attention mechanism
Liu and co-authors proposed a federated learning model with an attention mechanism to address the negative transferring problem due to the unreliable model parameters from the unlabeled clients. They performed the experiments on two datasets of fault diagnosis of rolling bearing and compared the model performance over seven models to justify the proposed model superiority on choices of using semi-supervised learning, federated learning strategies, and using of attention mechanism.
The manuscript is well prepared, with detailed background information and method description, the experiments are well designed, and the conclusions are well supported by the discussion. While as the attention mechanism is the merit of the model, it is better to have more investigation and have direct proof of the mechanism on the performance improvement. Overall, the work is of great interest to the machine learning community, and it deserves publications with the authors given an chance to consider the following comments.
a) The experiments using the second dataset by Shanghai Maritime University is more or less a replication of the ones from first dataset, the authors can simplify the discussion, and maybe focus more on the difference.
b) To directly show the impact of attention mechanism, it is better if the authors can consider showing the evolution of aggregation weights Pk_j and attention score Att_j over optimization of federation center parameters, with respect to clients with different portions of labelled and unlabeled data. The authors may have one or few more clients like client 2 but with different portions of labelled and unlabeled data. It would be expected that Att_j and Pk_j are higher for clients with more labelled data or higher accuracy, and the difference should decay during training.
c) In Table 2, the reference information is missing for Sem-Fed, or it needs more description for readers to understand the model.
d) In the discussion following Table 4, the comparison of FedSem and Sem-Fed needs more justification, partially due to the not detailed description of Sem-Fed.
Typos and formatting issues:
1: typo between Eq. 10 and Eq.11: “the experimental validation of this piper” -> paper
2: Table 5 for experiment 4, mean is not bold
3. Table 7, for client 2, the header should be unlabeled and unlabeled.
Author Response

(The authors gave the same response as above.)
